# An Axial Force Sensor Based on a Long-Period Fiber Grating with Dual-Peak Resonance

**Weixuan Luo** [1,2,†] **, Ying Wang** [3,†] **, Qiang Ling** [1,2] **, Zuguang Guan** [1,2] **, Daru Chen** [1,2] **and Qiong Wu** [2,4,*]

1   Hangzhou Institute of Advanced Studies, Zhejiang Normal University, 1108 Geng Wen Road, Hangzhou 311231, China; weixuanluo@zjnu.edu.cn (W.L.); qianglingoptics@zjnu.edu.cn (Q.L.); zgguan@zjnu.edu.cn (Z.G.); daru@zjnu.cn (D.C.)
2   Key Laboratory of Optical Information Detection and Display Technology of Zhejiang, Zhejiang Normal University, Jinhua 321004, China
3   Laboratory of Photo-Electric Functional Films, College of Science, University of Shanghai for Science and Technology, 516 Jun Gong Road, Shanghai 200093, China; yingwang_optics@st.usst.edu.cn
4   Institute of Information Optics, Zhejiang Normal University, Jinhua 321004, China
*   Correspondence: wuqiong@zjnu.cn
†   These authors contributed equally to this work and should be regarded as co-first authors.

**Abstract:** A high-sensitivity axial force sensor with a large measurement range based on a dual-peak long-period fiber grating (LPFG) is proposed and experimentally demonstrated. Previously, the relationship between the grating period and the dual-peak wavelengths has been investigated based on the coupled-mode theory. In our experiment, the LPFG was fabricated in our laboratory by illuminating the fiber core with the aid of a 213 nm UV laser. The sensitivity of the proposed axial force sensor can reach −14.047 nm/N in the force range from 0.490 N to 4.508 N. Taking the advantages of a compact size, low cost, and large measurement range, our force sensor has more applicable abilities in harsh environments.

**Keywords:** optical fiber sensors; long-period fiber grating; force





## 1. Introduction

Axial force measurements and control are of great importance in numerous industrial fields, nanoparticle interactions and human health monitoring [1–4]. A variety of axial force sensors have been proposed recently and can be divided into two categories: force-measuring platforms and installed miniature force sensors [5,6]. Installed miniature force sensors are widely used in the medical field because of their compact size. Among them, the optical fiber sensor is a typical miniature axial force sensor with the advantages of a small size, low cost, resistance to electromagnetic interference, and high response sensitivity.

Typical optical fiber sensors based on optical fiber interferometers and fiber grating have been reported to be used for force measuring [7]. Limited by a narrow free spectral range, optical fiber sensors based on interferometers have higher requirements for signal demodulation [8]. As the most popular optical fiber sensor structure, the traditional fiber Bragg gratings (FBGs) inscribed into single-mode fibers (SMFs) have a low axial force sensitivity of 44.116 pm/N [9]. To improve the axial force sensitivity, some researchers have proposed special FBG structures for axial force sensing. Koustav Dey et al. [10] proposed a half-etched FBG axial force sensor, where a maximum force sensitivity of 1.96 nm/N and measurement resolution of 0.01 N was achieved in the range of 0.2 N to 2.5 N. Wei Luo et al. [11] demonstrated the use of an ultra-high-sensitivity force FBG sensor written on 2.5 μm diameter microfiber. The force sensitivity can reach 3146 nm/N in the range of 0 N to 7.24 mN. Even though it exhibits an extremely high response sensitivity, its weak mechanical structure limits its practical application.

Long-period fiber gratings (LPFGs) allow for the co-propagating coupling of the core mode with several cladding modes [12]. It enables the LPFGs to have sensitivity to many

environmental parameters, such as temperature, strain, bending, torsion, surrounding refractive index (RI), and force [13]. For axial force measuring, a silica SMF-based LPFG with a force sensitivity of ~0.5 nm/N has higher sensitivity than that of a traditional FBG. Therefore, several techniques have been proposed for enhancing the axial force response, such as a fiber structure changing technique and writing an LPFG onto the special fibers. Xiaolan Li et al. [14] proposed a microbend-inserted LPFG for axial force sensing. The microbend was inserted at the edge of the LPFG by a $CO_2$ laser. A maximum force sensitivity of 41.24 nm/N can be achieved at the bending radius of 1.14 mm. A similar structure was used in a phase-shifted LPFG for force sensing [15]. Gabriela S. B. et al. [16] presented an LPFG fabricated in a polymer microstructure fiber using transverse periodic loading combined with fiber heating for force sensing. The results showed that the sensor has a linear response to force with a sensitivity of −1.39 nm/N. It is obvious that the above technologies can improve force sensitivity by damaging the mechanical structure of the optical fiber, but inevitably they make the optical fiber sensing head very fragile.

The dual-peak resonance for the high-order cladding modes in LPFGs has attracted attention because of its high RI sensitivity [17] and as it is extensively used in the biochemical sensing field with regard to viruses, drug ingredients, and serum albumin [18–20]. Many researchers have been devoted to the optimization of the RI sensitivity by certain methods (cladding etching and thin-film coating) [21]. However, the axial force sensor does not have the same rules as the RI sensor. Mechanical structural damage or fragile thin films are not suitable for carrying out actual force measurements in harsh environments.

In this work, an axial force sensor based on the dual-peak LPFG is proposed. The relationship between the grating period and the dual-peak wavelengths is discussed theoretically. The experimental setup and the force measuring methods are described. Finally, the experimental results of axial force sensing are detailed.

## 2. Working Principle

The schematic diagram of the axial force sensor based on the LPFG structure is shown in Figure 1a. The sensor is fabricated by illuminating the core with the aid of a 213 nm UV laser. This method of manufacturing gratings can obtain stable grating periods Λ and grating lengths L. To investigate the coupling mode and spectral characteristics of the LPFG axial force sensor, the coupling-mode theory was used to analyze the sensor's sensing characteristics. The transmission spectrum of the LPFG is usually obtained by the formula [22]:

$$T = |R(L)|^2 = \cos^2\left(\sqrt{\hat{\sigma}^2 + \kappa^2}L\right) + \frac{1}{1 + \kappa^2/\hat{\sigma}^2}\sin^2\left(\sqrt{\hat{\sigma}^2 + \kappa^2}L\right) \tag{1}$$

where $\hat{\sigma} = \delta + \frac{\sigma_{11} - \sigma_{22}}{2}$, and $\sigma_{11}$ is the self-coupling coefficient of the fiber core mode. $\sigma_{22}$ is the mutual coupling coefficient of the cladding mode. $\delta = \frac{1}{2}(\beta_{co} - \beta_{cl}) - \frac{\pi}{\Lambda} = 2\pi n_{eff}^{co}\left(\frac{1}{\lambda} - \frac{1}{\lambda_B}\right)$, where $\lambda_B$ is the central wavelength of the grating.

When the phase-matching condition is satisfied, the cladding mode is excited at the resonant wavelength and decays rapidly, resulting in a resonant peak, as shown by the red solid line in Figure 2a. The phase-matching condition for the LPFG can be expressed as [22]

$$\lambda_{res,m} = \left(n_{eff}^{co} - n_{eff}^{cl,m}\right)\Lambda \tag{2}$$

where $n_{eff}^{co}$ is the effective refractive index of the fiber core mode, $n_{eff}^{cl,m}$ is the effective refractive index of the cladding mode, and Λ is the grating period. In a certain wavelength range, the coupling between the fundamental and higher-order cladding modes can produce the dual peaks for the LPFG [18]. Figure 2 shows the phase-matching curves and transmission spectra of the 19th cladding mode. The phase-matching curve has only one intersection with a line of a 190.0 μm period, which is the phase-matching turning point (PMTP), and the LPFG operates at the PMTP with only one resonant peak. As the period decreases, the

single peak splits into dual peaks in opposite directions. When the period is reduced to 188.5 µm, the dual peaks shift to 1364 nm and 1654 nm. At this point, the LPFG operates in the dual-peak resonance near the PMTP and has a higher corresponding sensitivity to axial force. Therefore, a period of 188.5 µm was chosen to analyze the transmission spectrum of the LPFG.

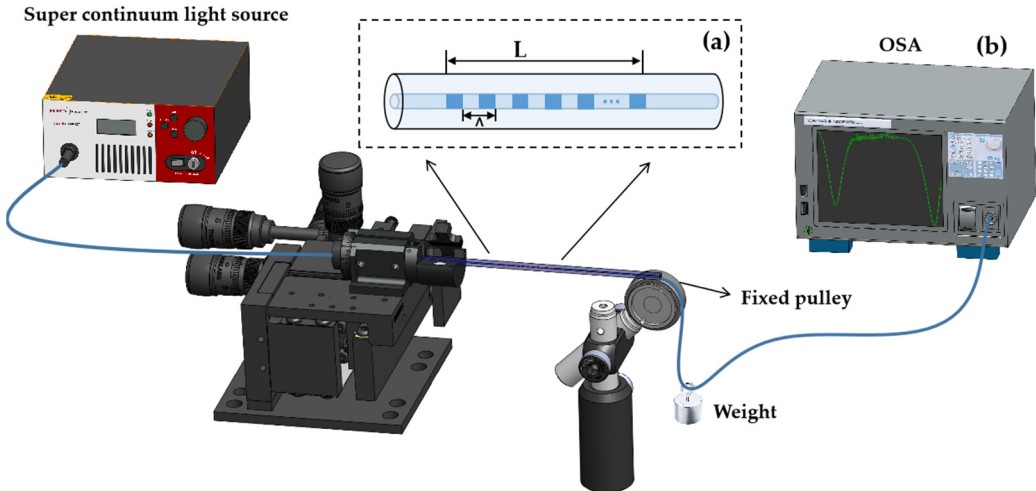

**Figure 1.** Schematic diagram of (**a**) fiber structure and (**b**) the experiment setup for axial force measurement.

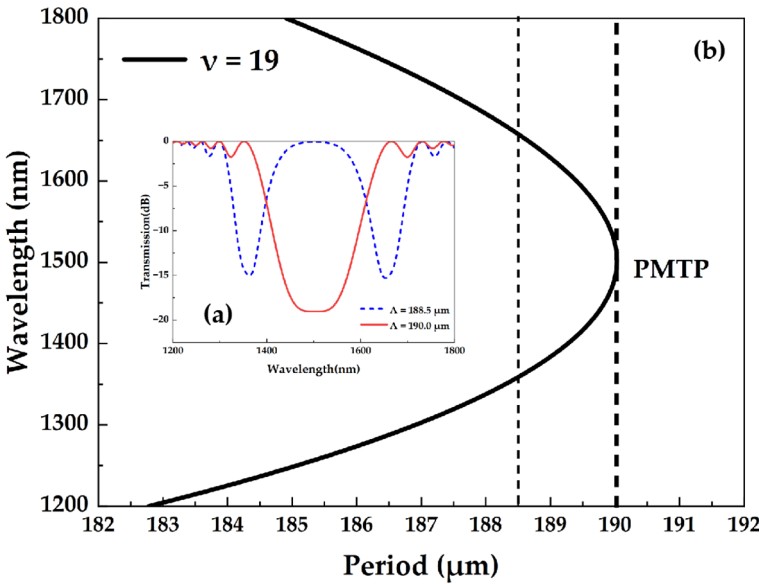

**Figure 2.** (**a**) Phase-matching curve and (**b**) transmission spectrum of the 19th cladding mode.

The position of the resonant peak is determined by the grating period, while the amount of loss in the resonant peak is determined by the grating length. Figure 3 shows the variation curve of transmittance with the grating length. It can be seen from the figure that the two resonant peaks can reach the full coupling state when the grating length is 1 cm, 3.1 cm, 5.2 cm, etc., and the central wavelength between the two peaks is at the position of 1.5 cm, 3 cm, 4.6 cm, etc., without loss. To ensure that there is no loss at the central wavelength between the two peaks and that the position of the two peaks is fully coupled as far as possible, a grating length of 3 cm is selected to analyze the LPFG transmission spectrum.

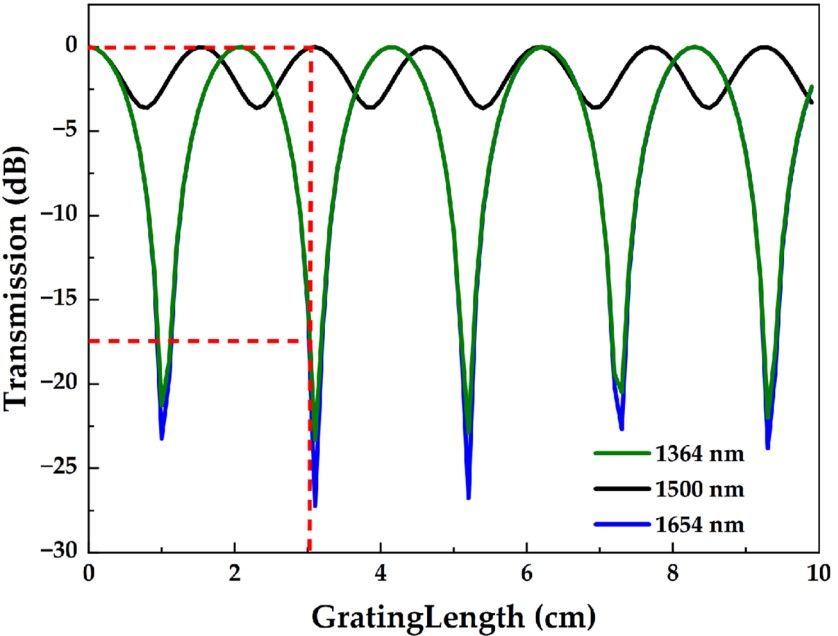

**Figure 3.** Variation in transmittance with grating length.

Based on the above theoretical analysis, the grating parameters used to simulate the transmission spectrum of the LPFG are as follows: The refractive indices of the core and cladding are 1.4681 and 1.4628, respectively. The core radius is 4.15 μm and the cladding radius is 62.5 μm. The average refractive index of the grating is $4 \times 10^{-4}$, the grating period is 188.5 μm, and the grating length is 3 cm. The transmission spectrum (black line) and energy distribution map (spot map) of the 19th cladding mode are obtained by simulation, as shown in Figure 4. The position of the dual peaks and the losses correspond to the results in Figures 2 and 3, indicating that the simulation results are correct. The above parameters will be used to analyze the LPFG axial force sensing.

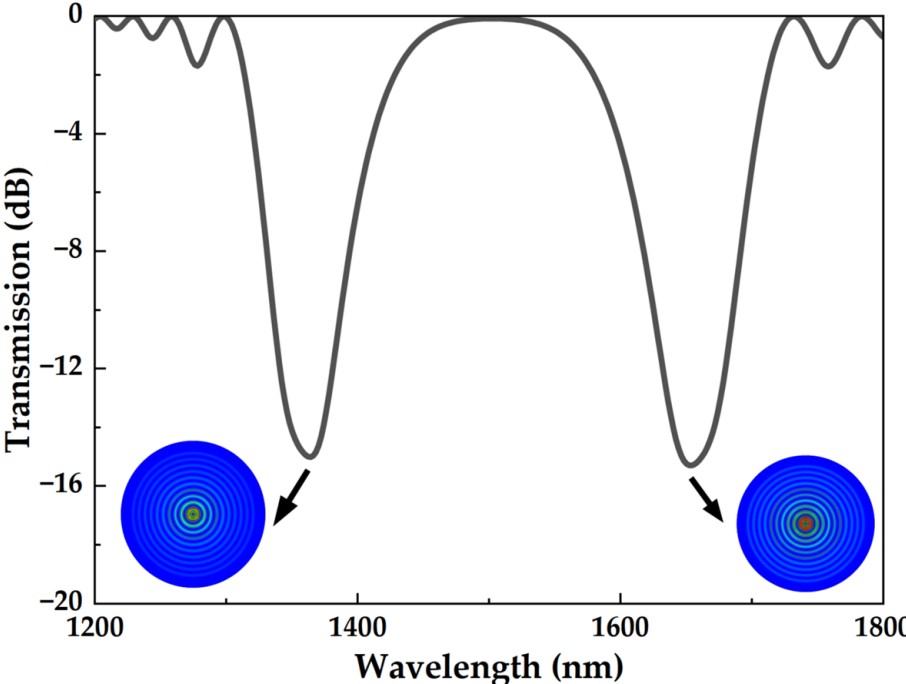

**Figure 4.** LPFG transmission spectrum and energy distribution of the dual-peak resonance near PMTP.

When the LPFG is subjected to axial forces, the grating period, fiber radius, core refractive index, and cladding refractive index all change, resulting in a drift in the grating resonant wavelength, which is the basic principle of axial force sensing in LPFGs [20]. With the axis of the fiber as the z-axis, the x- and y-axes are within the cross section of the fiber. Let us assume that the axial tension on the LPFG is F. The stress on the LPFG at this point is $\sigma_2$ = P (P = F/s, where s is the cross-sectional area of the fiber). According to Hooke's law, the strain in each direction can be expressed as [23]

$$\varepsilon = \begin{bmatrix} \varepsilon_x \\ \varepsilon_y \\ \varepsilon_z \end{bmatrix} = \begin{bmatrix} -\nu\varepsilon_z \\ -\nu\varepsilon_z \\ \varepsilon_z \end{bmatrix} = \begin{bmatrix} -\nu\frac{P}{E} \\ -\nu\frac{P}{E} \\ \frac{P}{E} \end{bmatrix} \tag{3}$$

where $E$ and $\nu$ are the Young's modulus and Poisson's ratio of the quartz fiber, respectively. For quartz fibers, Young's modulus $E$ is assumed to be 70 Gpa and Poisson's ratio $\nu$ is assumed to be 0.17. Therefore, after an axial force is applied to the LPFG, the grating period and fiber radius can be expressed as [24]

$$\Lambda' = \Lambda(1 + \varepsilon_z) = \Lambda\left(1 + \frac{P}{E}\right) \tag{4}$$

$$a'_i = a_i(1 + \varepsilon_r) = \Lambda\left(1 - \nu\frac{P}{E}\right) \tag{5}$$

where $i$ = 1,2 represents the core and the cladding of the fiber. In addition, the change in the transverse and longitudinal refractive indices of the core and cladding due to elastic effects can be expressed as [24]

$$n_{it} = n_i - \frac{1}{2}n_i(p_{12} - \nu(p_{11} + p_{12}))\varepsilon_z \tag{6}$$

$$n_{iz} = n_i - \frac{1}{2}n_i(p_{11} - 2\nu p_{12})\varepsilon_z \tag{7}$$

where $p_{11}$ and $p_{12}$ are the elongation coefficients (Pockels coefficients) of the fiber core and cladding, respectively. For quartz fibers, $p_{11}$ = 0.113 and $p_{12}$ = 0.252. In single-mode fibers, the longitudinal electric field is much smaller than the transverse electric field. Therefore, it is mainly the transverse refractive index $n_{it}$ that plays a role in the fiber core mode. Figure 5a–c simulates the relationship between the transmission spectrum, as well as the position of the dual peaks and their losses for different sizes of axial forces. It can be seen that as the force increases, the resonant peak at the short wavelength is red-shifted, while the resonance peak at the long wavelength is blue-shifted, i.e., the two peaks move closer and closer to the central wavelength. Additionally, the dual-peak loss decreases as the axial force increases. Figure 5d shows the scatter plot and fitted curve of the wavelength difference between the two resonance peaks. The sensitivity of this LPFG axial force sensor is given as −14.95 nm/N and the linearity of the fitted curve is 0.997, showing a good linear relationship.

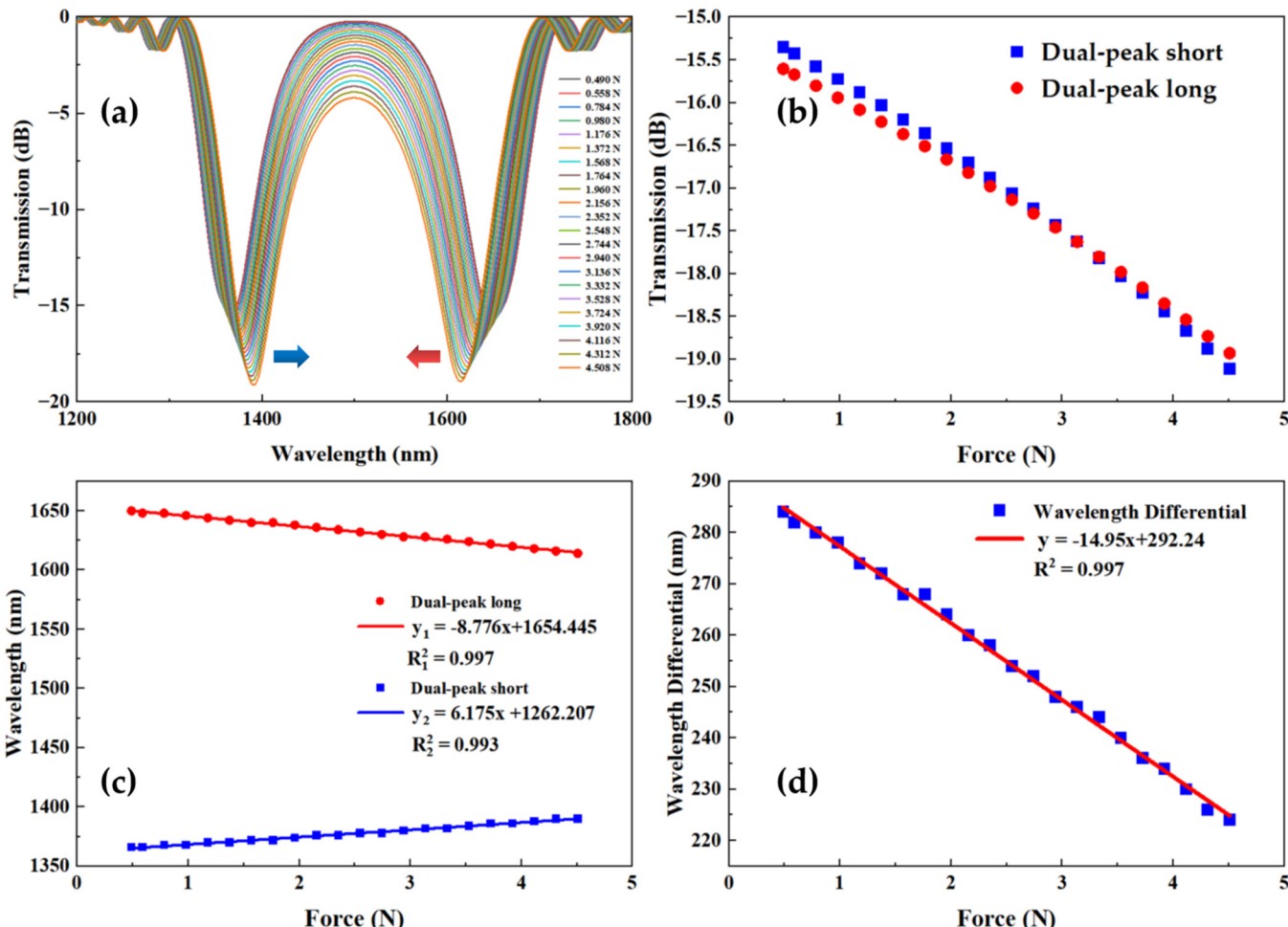

**Figure 5.** Simulation results; (**a**) Transmission spectrum of LPFG for different sizes of axial force; (**b**) the relationship between the variation in two resonant peak losses for different sizes of axial force; (**c**) relationships between the wavelengths of the two resonant peaks for different sizes of axial force; (**d**) scatter plot and fitted curve of the wavelength difference between the two resonant peaks for different sizes of axial force.

### 3. Experiments and Discussion

Based on the above simulation calculations, we manufactured a dual-peak LPFG axial force sensor with a grating period Λ of 185.9 um and a grating length L of 20.6 mm, which was fabricated by using the point-by-point inscription technique with a 213 nm ultraviolet laser (IMPRESSS 213, XITON). The pulse energy and exposure time were 135 mW and 15 s, respectively. The SMF (SM-28, Corning) used in the experiment before fabricating the LPFG was hydrogen-loaded in advance for two weeks at 120 °C and 12 MPa and was annealed at 100 °C for 12 h to improve thermal stability. Its spectral transmission is shown in Figure 6. The theoretical simulation results are similar to the experimental manufacturing results.

Figure 1b shows the schematic diagram of the experimental device for measuring the axial force of the proposed sensor. It includes a super-continuum light source (SuperK Compact, NKT Photonics), an OSA (AQ6375, Yokogawa), an optical fiber rotator (HFR007, Thorlabs), a fixed pulley, and a weight (composed of different quantities of weights). During the experiment, the sensor was slightly stretched, with one end fixed to the fiber optic rotator and the other end connected to a heavy object suspended at the center of the fiber optic through a fixed pulley. The fixed pulley was adjusted to the same height as the fiber optic rotator through the optical fixed seat, ensuring that the tensile direction of the fiber optic was axial. The broadband light from the super-continuum light source was connected to the sensor, and then the transmission spectrum of the sensor was recorded through

OSA. For the measurement of axial force, we represented the force based on the mass of the heavy object. By adding different masses of heavy objects, different axial forces were obtained. The mass of the heavy object was in the range of 0–460 g, which is approximately 0–4.508 N. Adding different masses of heavy objects results in different axial forces on the sensor, leading to varying degrees of shape change in the sensor.

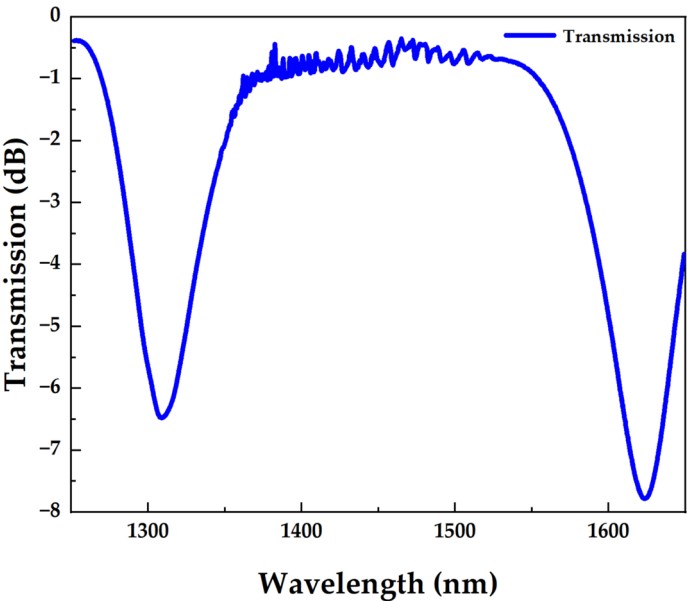

**Figure 6.** Dual-peak LPFG transmission spectrum.

Figure 7a shows the transmission spectra under different axial forces of the sensor. Additionally, as the force increases, the loss and wavelength of the resonant peak will change. Figure 7b,c shows the peak wavelength and the relationship between loss and force, respectively. It is shown that the short-wavelength resonant peak undergoes a red shift, while the long-wavelength resonant peak undergoes a blue shift, meaning that the two peaks are getting closer to the center wavelength. Additionally, the losses decrease with the increase in axial force. Therefore, the evidence shows that long-wavelength resonant peaks have a higher force sensitivity than short-wavelength resonant peaks, which is highly consistent with the results of our simulation. Figure 7d shows the scatter plot and fitting curve of the wavelength difference between the two resonant peaks. The high sensitivity of the LPFG axial force sensor is −14.047 nm/N, and the ultra-linearity of the fitted curve is 0.997, indicating a good linear relationship. This indicates that compared to general LPFG force sensors, the dual-peak LPFG has higher responsiveness and linearity with a small force.

Table 1 compares the axial force sensitivity values and the measurement range achieved in the present study with the different types of fiber gratings described in the literature. It is clearly indicated that the force sensitivity of our sensor is one order of magnitude higher than that of the half-etched FBG [10]. It is undeniable that using a microfiber to inscribe the FBG [11] can achieve a higher force sensitivity, but it also means that it has only a limited operating range and the sensor is extremely fragile. In the LPFG structure, our sensor exhibits good force sensitivity and a wide operating range. Compared with precision manufacturing [14,16], our structure is more compact, easy to manufacture, lower in cost, and more practical.

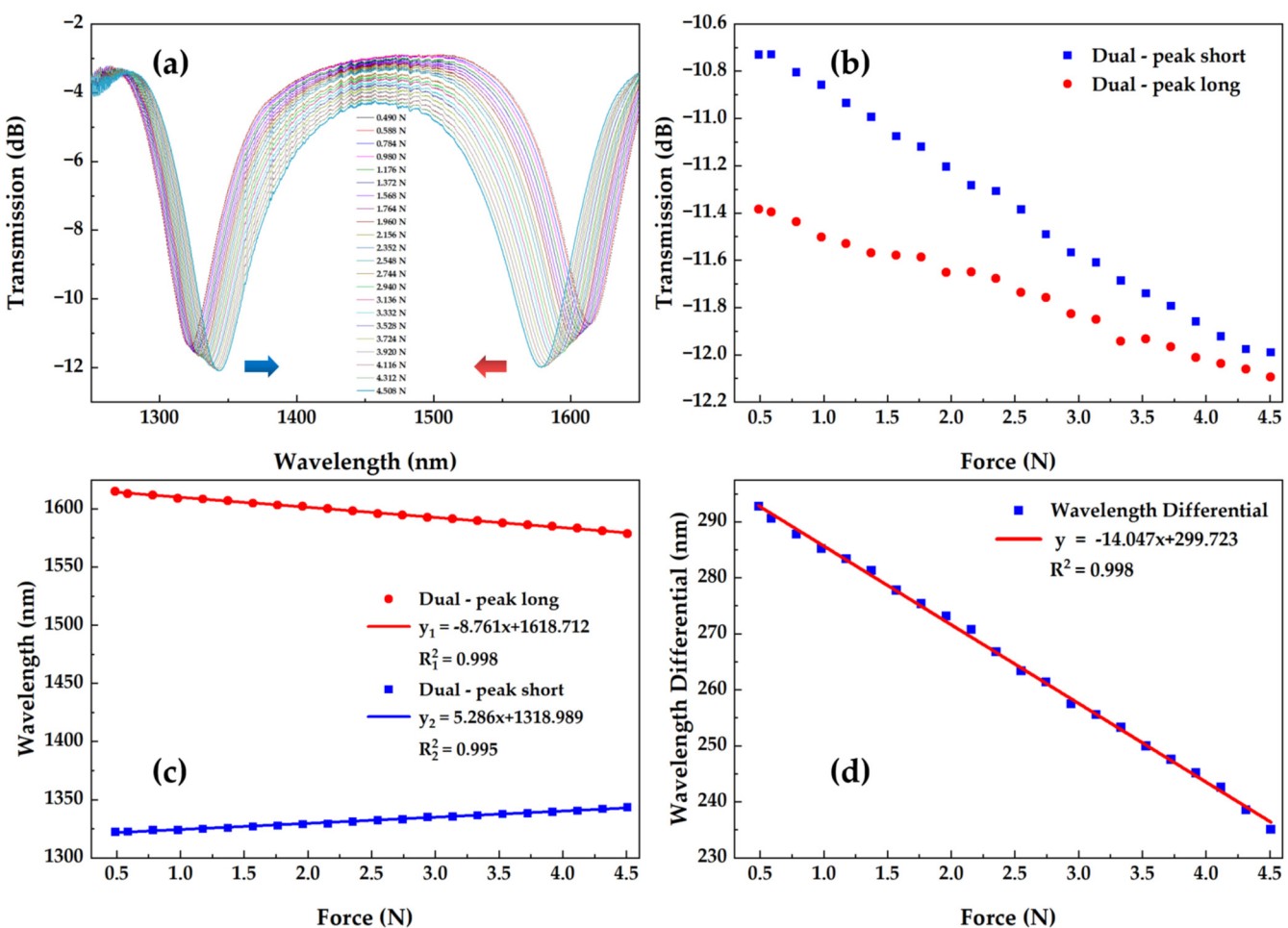

**Figure 7.** Experiment results; (**a**) Transmission spectrum of LPFG for different sizes of axial force; (**b**) the relationship between the variation in two resonant peak losses for different sizes of axial force; (**c**) relationships between the wavelengths of the two resonant peaks for different sizes of axial force; (**d**) scatter plot and fitted curve of the wavelength difference between the two resonant peaks for different sizes of axial force.

**Table 1.** Comparisons among the fiber grating-based sensors for axial force measurements.

| Structure | Axial Force Sensitivity | Range | Fabricated Method | Ref. |
|---|---|---|---|---|
| Half-etched FBG | 1.96 nm/N | 0.20–2.50 N | Chemical corrosion with hydrofluoric acid | [10] |
| Microfiber-tapered FBG | 3146 nm/N | 0–0.0062 N | Focused ion beam machining | [11] |
| Microbend LPFG | 41.24 nm/N | 0–1.90 N | Inserting a microbend at the edge | [14] |
| LPFG fabricated in a polymer microstructure fiber | 1.39 nm/N | 0–16 N | Transverse periodic loading combined with fiber heating | [16] |
| Dual-peak LPFG | 14.047 nm/N | 0.490–4.508 N | UV laser | Our work |

## 4. Conclusions

In summary, we have presented a high-sensitivity axial force sensor with a large measurement range based on the dual-peak LPFG and conducted theoretical simulation calculations on the coupling mode and spectral characteristics of the sensor. We also analyzed the relationship between the axial force and the dual-peak wavelength, and finally, conducted axial force experiment verification. Both our simulation and experimental results indicate that when the LPFG operates near the PMTP, it has high force sensitivity due to the dual-peak resonance. The experimental results show that the proposed axial force sensor has a high axial force sensitivity of −14.047 nm/N and a large operating range

from 0.490 N to 4.508 N. It has the advantages of high sensitivity, simple manufacturing, a compact structure, a large working range, and low losses, providing a new platform for manufacturing very simple and efficient axial force sensors.

**Author Contributions:** Conceptualization, W.L. and Q.L.; methodology, Q.L.; software, Y.W.; validation, W.L. and Y.W.; formal analysis, Z.G. and D.C.; investigation, W.L. and Y.W.; resources, W.L. and Q.L.; data curation, W.L. and Y.W.; writing—original draft preparation, W.L., Y.W. and Q.L.; writing—review and editing, D.C. and Q.W.; supervision, Q.W. All authors have read and agreed to the published version of the manuscript.

**Funding:** This research was funded by the "Pioneer" and "Leading Goose" R&D Program of Zhejiang (No. 2022C03084), the Zhejiang Provincial Natural Science Foundation of China (Nos. LQ22F050007 and LQ23F050004), and Ningbo Science and Technology Project (No. 2021Z030).

**Institutional Review Board Statement:** Not applicable.

**Informed Consent Statement:** Not applicable.

**Data Availability Statement:** Not applicable.

**Acknowledgments:** We thank the Hangzhou Institute of Advanced Studies of Zhejiang Normal University for the scientific help and support throughout this research.

**Conflicts of Interest:** The authors declare no conflict of interest.

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
