# Peer review of "An Axial Force Sensor Based on a Long-Period Fiber Grating with Dual-Peak Resonance"

_photonics, doi:10.3390/photonics10050591_

Round 1

Reviewer 1 Report

In this manuscript, the authors propose an all-fiber sensor for axial force measurement. The sensors exhibits a high axial force sensitivity and a large measurement range. In my view, the results of the presenting work are interesting, innovative and have a valuable contribution to the design of axial force fiber sensors community and related fields. This article can be accepted after minor revision.

1.What do the figures represent for in Figs.4? Please indicate that.

2.The temperature plays an important role in the properties of the loss transmission in the LPFG. How is temperature considered in the work? Please clarify.

3.The advantages of “dual-peak” sensing over “single peak” sensing is not well reflected, please add data or text to clearly explain.

4.The References for Eq. 3-7 need to be provided

5.The authors should provide some information about the setting of the theoretical simulation calculations on the coupling mode and spectral characteristics, so the reviewers can judge the appropriateness of the simulation, and the readers can follow their work if interested.

Author Response

Point 1: What do the figures represent for in Figs.4? Please indicate that.

Response 1: Thanks for the valuable comments. The light spot in Figure 4 is the energy distribution of the 19th cladding mode. We have added some statements on this issue in the revised manuscript.

“The transmission spectrum (black line) and energy distribution map (spot map) of the 19th cladding mode was obtained by simulation, as shown in Figure 4.”

(Line 126-128, Page 4)

Point 2: The temperature plays an important role in the properties of the loss transmission in the LPFG. How is temperature considered in the work? Please clarify.

Response 2: Thanks for the valuable comments. In the axial force sensing experiment, a room temperature of 26 ℃ was selected. For the thermostatic chamber, the temperature fluctuation in the thermostatic experiment is 0.1 ℃ within 1 min of the experimental time, and the wavelength shift of the bimodal LPFG is about 0.34 nm [1]. Based on the sensitivity of the LPFG axial force sensor at -14.95 nm/N near the PMTP, the axial force deviation caused by temperature fluctuations can be estimated to be about 0.02 N. The effect of temperature fluctuations is negligible. This paper mainly focuses on the measurement of axial force, temperature measurement is not the focus of this paper. In addition, several types of temperature cross-sensitivity problems suitable for bimodal resonant LPFG have been reported, such as desensitizing package [2], double demodulation [3], etc. In a word, temperature is not the focus of this paper, so it will not be repeated.

[1] Xuewen, S.; Lin, Z.; I. Bennion. High sensitivity sensors utilising characteristics of dispersion-turning-point of long-period gratings in B/Ge co-doped fibre. in 15th Optical Fiber Sensors Conference Technical Digest (2002), Vol. 1, pp. 573– 576. doi:10.1109/OFS.2002.1000806.

[2] Y. Zhang.; Bai-Ou, G.; Xinyong, D. A novel fiber grating sensor for simultaneous measurement of strain and temperature based on prestrain technique, Chin. J. Lasers 28 (8) (2001) 729–731.

[3] Ting, L.; Yaowei, L. Xinyi, D. Wenbo, G.; Xunsi, W. Shaocong, D.; Baoan, S.; Tiefeng, X. Peiqing, Z. Simultaneous detection of temperature, strain, refractive index, and pH based on a phase-shifted long-period fiber grating. J LIGHTWAVE TECHNOL, doi: 10.1109/JLT.2023.3254550.

Point 3: The advantages of “dual-peak” sensing over “single peak” sensing is not well reflected, please add data or text to clearly explain.

Response 3: Thanks for the valuable comments. The axial sensing sensitivity results of theoretical simulation are shown in Figure 5c and Figure 5d. The maximum axial force sensing sensitivity calculated using only "single peak" is -8.776 nm/N, while the axial force sensing sensitivity calculated using "double peak" is -14.95 nm/N. From the experimental results, Figure 7c and Figure 7d indicate that the maximum axial force sensing sensitivity calculated using "single peak" is -8.761 nm/N, while the axial force sensing sensitivity calculated using "double peak" is -14.047 nm/N. The theoretical simulation results are similar to the experimental results, both clearly indicating that "bimodal" sensing has better axial force sensing sensitivity than "single peak" sensing.

Point 4: The References for Eq. 3-7 need to be provided.

Response 4: Thanks for the valuable comments. We have added the references to the text according to the reviewer's comments.

“Therefore, after an axial force is applied to the LPFG, grating period and fiber radius can be expressed as [23]”

(Line 144, Page 5)

“In addition, the change in the transverse and longitudinal refractive indices of the core and cladding due to elastic effects can be expressed as [23].”

(Line 145-147, Page 5)

Point 5: The authors should provide some information about the setting of the theoretical simulation calculations on the coupling mode and spectral characteristics, so the reviewers can judge the appropriateness of the simulation, and the readers can follow their work if interested

Response 5: Thanks for the valuable comments. We have provided the parameters used for the theoretical simulations in the second chapter of the manuscript. See the original text for details.

(Line 120-126, Page 4)

Reviewer 2 Report

The authors propose an axial force sensor based on long period fiber grating, which is experimentally validated based on theoretical simulations. The structure is more compact and easier to make than those in other articles, is more innovative and deserves to be published, but there are several problems:

1. "the grating length of 3 cm is selected to analyze the LPFG transmission spectrum" is mentioned in the article. However, in the later content, "the grating length is 3.1cm" is adopted. The parameters before and after are inconsistent. Is it written incorrectly? In addition, there are some language issues suggested to change.

2. The period and length of the grating used in the simulation were 188.5 μm and 3 cm respectively, while the period and length of the grating used in the experiment were 185.9 μm and 20.6 mm respectively. Why is there a difference between experiment and theory?

3. Does temperature affect this sensor? If present, how should it be considered?

4. There is lack of real application for the axial force sensor.

none

Author Response

Response to Reviewer 2 Comments

Point 1: "the grating length of 3 cm is selected to analyze the LPFG transmission spectrum" is mentioned in the article. However, in the later content, "the grating length is 3.1cm" is adopted. The parameters before and after are inconsistent. Is it written incorrectly? In addition, there are some language issues suggested to change.

Response 1: Thanks for the valuable comments. For the raster length, we have made corrections in the original text. We have also revised the grammar and expressions in the full text. See the original text for details.

(Line 126, Page 4; Line 144-147, Page 5)

Point 2: The period and length of the grating used in the simulation were 188.5 um and 3 cm respectively, while the period and length of the grating used in the experiment were 185.9 um and 20.6 mm respectively. Why is there a difference between experiment and theory?

Response 2: Thanks for the valuable comments. The difference between the period and length of the grating used in theory and experiment is due to the uncontrollable amount of variation in the average refractive index of the core. Eq.  is given in the manuscript [1]. The amount of change in the average refractive index of the core can cause a shift in the central wavelength. And the central wavelength is related to the grating period as follows: . The two equations above therefore show that the amount of change in the average refractive index of the core affects the selection of the grating period. In the theoretical simulation section, the average refractive index of the core is determined to be 4×10-4, which corresponds to a grating period of 188.5 um. However, during the experiments we were unable to control the average refractive index variation of the cores to an exact amount of 4×10-4. Therefore, the grating period used in theoretical simulation is inconsistent with that used in experiment. If the grating length in the theoretical simulation is used, the grating cannot reach the full coupling state. Therefore, the grating length should also vary with the grating period.

[1] Erdogan, T. Fiber grating spectra. J LIGHTWAVE TECHNOL 1997, 15(8): 1277-1294. doi:10.1109/50.618322.

Point 3: Does temperature affect this sensor? If present, how should it be considered?

Response 3: Thanks for the valuable comments. In the axial force sensing experiment, room temperature 26 ℃ was selected. For the thermostatic chamber, the temperature fluctuation in the thermostatic experiment is 0.1 ℃ within 1 min of the experimental time, and the wavelength shift of the bimodal LPFG is about 0.34 nm [2]. Based on the sensitivity of the LPFG axial force sensor at -14.95 nm/N near the PMTP, the axial force deviation caused by temperature fluctuations can be estimated to be about 0.02 N. The effect of temperature fluctuations is negligible. This paper mainly focuses on the measurement of axial force, temperature measurement is not the focus of this paper. In addition, several types of temperature cross-sensitivity problems suitable for bimodal resonant LPFG have been reported, such as desensitizing package [3], double demodulation [4], etc. In a word, temperature is not the focus of this paper, so it will not be repeated.

[2] Xuewen, S.; Lin, Z.; I. Bennion. High sensitivity sensors utilising characteristics of dispersion-turning-point of long-period gratings in B/Ge co-doped fibre. in 15th Optical Fiber Sensors Conference Technical Digest (2002), Vol. 1, pp. 573– 576. doi:10.1109/OFS.2002.1000806.

[3] Y. Zhang.; Bai-Ou, G.; Xinyong, D. A novel fiber grating sensor for simultaneous measurement of strain and temperature based on prestrain technique, Chin. J. Lasers 28 (8) (2001) 729–731.

[4] Ting, L.; Yaowei, L. Xinyi, D. Wenbo, G.; Xunsi, W. Shaocong, D.; Baoan, S.; Tiefeng, X. Peiqing, Z. Simultaneous detection of temperature, strain, refractive index, and pH based on a phase-shifted long-period fiber grating. J LIGHTWAVE TECHNOL, doi: 10.1109/JLT.2023.3254550.

Point 4: There is lack of real application for the axial force sensor.

Response 4: Thanks for the valuable comments. Micromanipulation and biological, material science, and medical applications often require control or measure the forces asserted on small objects. Limited accuracy or lack of force feedback has become a key challenge for surgeons to accurately perceive surgical instrument tissue interactions. For example, extensive surveys show that more than 50% of surgical errors are attributed to excessive or insufficient force input in laparoscopic cholecystectomy, which is one of the important factors affecting the results of MIS operations. The solution of forced feedback can enable surgeons to perceive accurate interactions, explore tissue characteristics, identify the anatomical structure of lumps or tumors during tissue palpation in MIS, improve surgical quality, and develop advanced robot-related technologies for high-level surgical autonomy. Due to the outstanding inherent advantages of fiber sensors, including micro size, insensitivity to electromagnetic interference, strong corrosion resistance and biocompatibility in the human internal environment, and insulation performance caused by electrical disconnection, fiber optic sensors have great potential for application in MIS surgery [5].

[5] Zhongxin, T.; Shunxin, W.; Ming, L.; Chaoyang, S. Development of a Distal Tri-Axial Force Sensor for Minimally Invasive Surgical Palpation. IEEE Trans. Med. Robot. Bionics 2022, 145-155. doi:10.1109/TMRB.2022.3142361.

Reviewer 3 Report

This work presents the design and implementation of an axial force tensor based on an LPFG with a dual peak resonance. I believe the paper is clearly explained and easy to understand, however, I suggest adding the following useful information:

1. A greater detail on the simulation may be useful, it was implemented with commercial software?  

2. In the first paragraph of section 3 the authors wrote: 

    ... The theoretical and simulation results are similar to the experimental manufacturing results.

I suggest adding the experimental values of the two wavelength peaks and compare with the theoretical values.

3.  In the second paragraph of section 3, the authors refer to heavy objects used to vary the axial force. I believe that a precise definition of the nature of the heavy object is needed.

In some parts, the acronym LPFG is written as LPEG.

In the third paragraph of section 3, the authors refer to Fig. 8d  as a scattering plot, it may be better to use the same text as in the figure, i.e. scatter plot.

Author Response

Point 1: A greater detail on the simulation may be useful, it was implemented with commercial software?

Response 1: Thanks for the valuable comments. In this manuscript, MATLAB is the software used for theoretical simulation. And we have provided the parameters used for the theoretical simulations in the text in accordance with the reviewers' comments. See the original text for details.

(Line 122-126, Page 4)

Point 2: In the first paragraph of section 3 the authors wrote:

    ... The theoretical and simulation results are similar to the experimental manufacturing results.

I suggest adding the experimental values of the two wavelength peaks and compare with the theoretical values.

Response 2: Thanks for the valuable comments. In the theoretical simulation, we used simulation parameters: an average refractive index change of 4x10-4 for the fiber core, a grating period of 188.5 um and a grating length of 3 cm. We obtained the positions of the two peaks at 1362 nm and 1654 nm. The grating period of the sensor we manufactured in the experiment is 185.9 um, and the grating length is 20.6 mm. The change in the average refractive index of the fiber core will have an impact on the selection of grating period. However, during the experimental process, we were unable to control the change in the average refractive index of the fiber core. Therefore, the grating period used in theoretical simulation is inconsistent with the grating period and grating length used in the experiment. Finally, the dual-peak positions obtained in the experiment are 1308 nm and 1624 nm. We can calculate that the theoretical simulation results in a dual-peak wavelength difference of 292nm, while our experimental results show a dual-peak wavelength difference of 316nm, indicating a difference of only 24nm. Meanwhile, from the axial force test results, our sensor also showed high consistency with the results obtained from theoretical simulation.

Point 3: In the second paragraph of section 3, the authors refer to heavy objects used to vary the axial force. I believe that a precise definition of the nature of the heavy object is needed.

Response 3: Thanks for the valuable comments. The weight is mainly achieved by using different quantities of standard weights. The weights are placed in a container and fixed together with the fiber with a thin rope. And the container naturally vertically downwards to ensure that the direction of force on the fiber is in the axial direction. Therefore, increasing the amounts of weights can easily change the axial force of the sensor. We have added some descriptions of Weight in the manuscript.

“...a fixed pulley and a weight (composed of different quantities of weights).”

(Line 182-183, Page 7)

Point 4: In some parts, the acronym LPFG is written as LPEG.

Response 4: Thanks for the valuable comments. We have revised the expression of the full text . See the original text for details.

(Line 168, Page 6; Line 206, Page 7)

Point 5: In the third paragraph of section 3, the authors refer to Fig. 8d  as a scattering plot, it may be better to use the same text as in the figure, i.e. scatter plot.

Response 5: Thanks for the valuable comments. We have modified all the expression in the revised manuscript according to the reviewer’s comments.

(Line 202, Page 7)
